# Overcoming Hypoxia-Induced Chemoresistance in Cancer Using a Novel Glycoconjugate of Methotrexate

**DOI:** 10.3390/ph14010013

**Published:** 2020-12-24

**Authors:** Marta Woźniak, Gabriela Pastuch-Gawołek, Sebastian Makuch, Jerzy Wiśniewski, Piotr Ziółkowski, Wiesław Szeja, Monika Krawczyk, Siddarth Agrawal

**Affiliations:** 1Department of Pathology, Wrocław Medical University, Marcinkowskiego 1, 50-368 Wrocław, Poland; marta.wozniak@umed.wroc.pl (M.W.); sebastian.mk21@gmail.com (S.M.); piotr.ziolkowski@umed.wroc.pl (P.Z.); 2Department of Organic Chemistry, Bioorganic Chemistry and Biotechnology, Faculty of Chemistry, Silesian University of Technology, Krzywoustego 4, 44-100 Gliwice, Poland; gabriela.pastuch@polsl.pl (G.P.-G.); wieslaw.szeja@adres.pl (W.S.); 3Biotechnology Centre, Silesian University of Technology, Krzywoustego 4, 44-100 Gliwice, Poland; 4Department of Medical Biochemistry, Wroclaw Medical University, Marcinkowskiego 1, 50-368 Wrocław, Poland; jerzy.wisniewski@umed.wroc.pl; 5Department and Clinic of Internal Medicine, Occupational Diseases, Hypertension and Clinical Oncology, Wroclaw Medical University, Marcinkowskiego 1, 50-368 Wrocław, Poland

**Keywords:** glycoconjugates, tumor microenvironment, methotrexate conjugate, hypoxia, cancer

## Abstract

The oxygen and nutrient-deprived tumor microenvironment is considered a key mechanism responsible for cancer resistance to chemotherapy. Methotrexate (MTX) is a widely incorporated chemotherapeutic agent employed in the treatment of several malignancies. However, drug resistance and systemic toxicity limit the curative effect in most cases. The present work aimed to design, synthesize, and biologically evaluate a novel glucose-methotrexate conjugate (Glu-MTX). Our study showed that Glu-MTX exerts an increased cytotoxic effect on cancer cells in comparison to MTX in hypoxia (1% O_2_) and glucose starvation conditions. Furthermore, Glu-MTX was found to inhibit the proliferation and migration of cancer cells more effectively than MTX does. Our results demonstrate that the conjugation of MTX to glucose led to an increase in potency against malignant cells under oxygen and nutrient stress. The observations shed light on a potential therapeutic approach to overcome chemoresistance in cancer.

## 1. Introduction

Chemotherapy is the leading treatment modality in oncological care and is commonly applied in combination with surgery or radiotherapy, depending on tumor advancement. During cancer progression, the tumor increases in size and triggers a series of events, including hypoxia and a nutrient-deficient environment [1]. These responses arise from the increase in oxygen and nutrient consumption due to significant growth of malignant proliferation, as well as an inadequate supply of substrates to the cells due to the formation of an irregular tumor microvasculature with leaky vessels [2]. An oxygen- and nutrient-deprived tumor microenvironment is considered a key mechanism responsible for cancer resistance to current treatment modalities, including chemotherapy, radiotherapy, and photodynamic therapy [3]. The critical role of hypoxia in chemotherapy resistance is well-documented and involves several pathways, predominantly the upregulation of hypoxia-inducible factor-1α (HIF-1α). The HIF-1α mediates the angiogenesis, invasion, metastasis of malignant cells; induces glucose transporters (GLUT) to increase glucose import; and contributes to chemotherapy resistance by enhancing the expression of membrane efflux pump P-glycoprotein (P-gp), which identifies chemotherapeutic drugs and removes them from cells [4]. These events are among the primary contributors to multidrug resistance, which often results in cancer relapse and higher mortality [5]. Thus, breaking hypoxia-induced drug resistance is necessary to elevate the efficacy of cancer chemotherapy and increase the patient’s lifespan. 

To survive in an oxygen- and nutrient-deprived environment and address the energy demands resulting from rapid proliferation, tumor cells significantly increase glucose uptake and the flux of metabolites through glycolysis [6]. This metabolic shift, termed “the Warburg effect”, is one of cancer’s most common traits and provides clinically corroborated strategies for cancer diagnostics and treatment. 

Chemotherapy is the leading treatment modality in oncological care and is commonly applied in combination with surgery or radiotherapy, depending on tumor advancement. The modification of biologically active compounds with polymers is one way to alter and control their pharmacokinetics, biodistribution, and often toxicity [7]. The primary underlying mechanism proposed for nanomedicine-based cancer therapy is passive targeting associated with enhanced permeability and retention [8]. To be most effective, anticancer drugs must penetrate tissue efficiently, reaching all the cancer cells that comprise the target population in a concentration sufficient to exert a therapeutic effect. The therapeutic effect of conjugates of biologically active compounds with polymers is reduced because of limited penetration [9].

Based on the overexpression of specific receptors on tumor cells, ligand-targeted drug delivery has been developed with the ability to efficiently deliver imaging agents in the tumor area or drugs into tumor cells via receptor-mediated endocytosis [10]. As mentioned above, elevated glucose intake and GLUT overexpression frequently occur in neoplasms and provide clinically corroborated strategies for cancer treatment [11,12,13]. Therefore, glycoconjugation, in which known cytotoxins or targeted anticancer therapeutics have been linked to glucose to improve cancer targeting and cellular selectivity, has become an appealing strategy for the targeted delivery of anticancer drugs [14,15]. These strategies employ the use of conjugates of d-glucose and bioactive molecule methotrexate (MTX) to improve efficacy and lower toxicity. MTX has been successfully used for many years in the treatment of patients with cancer and as an anti-inflammatory drug for the treatment of inflammatory diseases, such as rheumatoid arthritis.

There are many novel delivery systems that have been developed to improve the pitfalls of MTX therapy, ranging from polymeric conjugates, such as human serum albumin, liposomes, microspheres, solid lipid nanoparticles, polymeric nanoparticles, dendrimers, polymeric micelles, in situ forming hydrogels, and carrier erythrocyte, to nanotechnology-based vehicles such as carbon nanotubes, magnetic nanoparticles, and gold nanoparticles. Some of them are further modified with targeting ligands for active targeting purposes [16]. The pharmacokinetic properties of MTX polymeric conjugates are unsatisfactory because of their low penetration into cancer cells. 

In the present work, we designed, synthesized, and biologically evaluated a novel glucose-methotrexate conjugate (Glu-MTX). Here, we investigated whether Glu-MTX could overcome MTX chemoresistance in oxygen and glucose-deprived cancer cells and the relative molecular mechanisms. This research aimed to assess the possibility of overcoming tumor microenvironment-induced drug resistance by conjugating a chemotherapeutic agent to glucose.

## 2. Results

### 2.1. Chemistry

When designing the synthesis of glycoconjugate derivatives of MTX, the following assumptions were made: glycoconjugate is selectively transferred to the cancer cell by GLUT proteins responsible for the transfer of d-glucose to tumor cells. To increase the water solubility of the prodrug and increase the affinity for GLUT transmembrane proteins, the conjugate contains two sugar units. The designed construct contains d-glucose linked via a linker to MTX. The key bonds connecting both molecules are susceptible to the action of hydrolytic enzymes, which allows the release of MTX, d-glucose, and the linker in the cell. d-glucose is connected to the linker via a β-glycosidic bond that is susceptible to hydrolysis catalyzed by glycoside hydrolases. On the other hand, MTX is connected to the linker by a carbamate bond formed with the participation of amino groups. The cleavable linkage allows the release of the cytotoxic payload inside the malignant cells, possibly through enzymatic hydrolysis. The critical step of connecting the d-glucose derivative to MTX is accomplished in the 1,3-dipolar cycloaddition of the azide to a terminal alkyne bond in a variant developed by Sharpless [17], which is a method widely used in the synthesis of biologically active compounds [18]. 

The ability of substituted glucose analogs to be substrates for GLUT-1 has been investigated [19,20]. Kinetic and computational modeling studies using glucose analogs suggest that the hydroxyl groups at positions 1 and 3 and the pyran oxygen in the d-glucose most thermodynamically stable conformation are involved in stabilizing hydrogen bonding interactions with amino acid residues within the transporter. The loss of hydrogen bond acceptors at these positions makes glucose analogs poor substrates for GLUT-1. Thus, these data suggest that for glucose conjugates to remain substrates for GLUT-1, compounds with hydrogen bond acceptors such as nitrogen or oxygen must be retained proximal to carbons 1 and 3, and substitutions at the C1 position may retain a higher affinity for GLUT-1 if they are present in an equatorial conformation. A large number of known glucose conjugates are conjugated to the anticancer agent at position 1, with the C1 oxygen intact and locked into the equatorial β-d-position [21,22]. 

Based on literature data, a synthesis of glycoconjugate was designed in which d-glucose via a linker is associated with the cytotoxic compound methotrexate (Figure 1).

The sugar unit occurring in the most thermodynamically stable ^4^C_1_ conformation is connected by the β-*O*-glycosyl bond with the spacer because, as is evident from the literature studies, such an orientation is preferred by the GLUT-1 transporting protein. The next fragment of the spacer contains the 1,2,3-triazine system because, as can be expected from the literature studies, the introduction of a functional group capable of hydrogen bonding increases the affinity of the compound to the transporting protein. The methotrexate is connected to the linker by forming a carbamate bond. After the introduction into the cell, the designed conjugate is susceptible to hydrolysis catalyzed by hydrolytic enzymes (glucosidases, peptidases) overexpressed in cancer cells following the release of a cytotoxic substance in the cancer target.

The coupling of two molecular components with different properties promises to generate a new conjugate with unprecedented biological activity, as different molecular segments can act together [23]. This perspective is a new, practically simple, and very reliable fast-growing approach for the development of pharmaceutically important drug-like molecules that can accelerate drug discovery research for human use. Sharpless et al. [17] discussed the Cu(I)-catalyzed azide−alkyne 1,3-dipolar cycloaddition (“click chemistry”), a set of powerful, selective, and reliable reactions for coupling molecular fragments under mild reaction conditions. The option to combine bioconjugation with click chemistry has emerged as a versatile tool with a wide range of applications. Effectively, the 1,2,3-triazole ring results in an ideal linker in bioconjugation, as (a) it presents a good water solubility, thus allowing in vivo administration; (b) it is analogous to an amide function for its electronic properties but is resistant to hydrolysis; (c) it is sufficiently stable in biological systems; and, finally, (d) it is a rigid linker, which allows internal interaction between the two linked moieties to be avoided. The unique features of click chemistry provide a toolbox for efficient coupling methodologies for the synthesis of a variety of conjugates [18,24]. Of the click reactions that have been developed, the most widely applied is the copper-catalyzed azide−alkyne cycloaddition reaction (CuAAC). Considering the advantages of this solution in our glycoconjugate synthesis project, a sugar unit containing a terminal azide group was combined with a propargyl carbamate-derived methotrexate. This convenient approach enables the rapid synthesis of carbohydrate conjugates in which the heterocyclic triazole ring serves as a shackle for joining the carbohydrate moiety to the biomolecule. When we move toward carbohydrate chemistry, the sugar moiety can be easily furnished with an azide functionality with routine synthetic protocols [25,26]. One of the substrates in the synthesis of glycoconjugate was the 2-azidoethyl β-d-glucopyranoside **1**. It was obtained as β-glucoside in a coupling reaction between 2-bromoethanol and 1,2,3,4,6-penta-*O*-acetyl-β-d-glucose in the presence of boron trifluoride diethyl etherate (BF_3_·Et_2_O) as a catalyst [26]. An azido function was introduced via the S_N_2 displacement of the bromine, using sodium azide in *N,N*-dimethylformamide (DMF) [25]. The last step was deprotection of the *O*-acetyl-protected glucoside under Zemplén conditions by the use of sodium methoxide in MeOH [25,27].

The second intermediate substrate is a derivative of methotrexate bearing a terminal acetylenic group. This synthesis was much more challenging. Such a group can be attached to methotrexate via an amide or a carbamate bond. Methotrexate is an unstable compound that is practically insoluble in organic solvents applied in the amidation reaction. A more convenient way to prepare amides could be the direct condensation of carboxylic acids and amines. Nevertheless, it is known that such an “ideal” amidation process needs very harsh conditions (temperature above 100 °C) to circumvent unreactive carboxylate-ammonium salts formation toward the desired amide bond formation [28]. This is adverse because other sensitive functionalities are present within coupled compounds. Therefore, the activation of carboxylic acid seems to be necessary [28]. The results of the conducted experiments associated with the selection of activation conditions are presented in Table 1.

There are numerous commercially available coupling reagents for constructing an amide bond, including carbodiimides alone or plus additives such as HOBt or DMAP [29]. The carbodiimide reacts with the carboxylic acid to form *O*-acylisourea mixed anhydride, which can react directly with an amine to yield the desired amide. We applied the DCC/DMAP condensing system for coupling propiolic acid with methotrexate. The reaction was carried out at room temperature for 48 h in different solvents. Unfortunately, the complex reaction mixture was formed, and the desired product was isolated in a poor yield. The low yield of the desired product induced us to search for another method of MTX functionalization. The reaction of acyl halides with an amino group is, in principle, the simplest approach to amide bond synthesis. Propargyl acid chlorides are unstable, and we choose the commercially available propargyl chloroformate. Such a reaction requires the presence of a tertiary amine in the reaction medium. A range of tertiary amines was tested; however, the reaction of MTX with propargyl chloroformate in the presence of a variety of organic bases such as pyridine, Et_3_N, DMAP, and imidazole in several different solvents gave MTX derivatives in very low yields. The treatment of methotrexate with two equivalents of propargyl chloroformate and tertiary amine in DMF/CH_2_Cl_2_ at room temperature gave the mixed anhydrides [29], which were condensed with amine groups of MTX, and an inseparable reaction mixture was obtained. An extremely effective acylating agent, 1-carboxybenzyl 3-ethylimidazol triethyloxonium tetrafluoroborate, was applied in the synthesis of nucleoside carbamates [30]. Assuming that an analogous acylating agent can be formed in a reaction of propargyl chloroformate **2** and *N*-methylimidazole, this compound was selected in subsequent reactions. When *N*-methylimidazole (NMI) was used, the yield of the desired carbamate derivative of MTX **4** was significantly improved depending on the solvent (DMF, CH_2_Cl_2_, CH_3_CN, THF, CHCl_3_). The optimal yield of **4** was obtained in the reaction with four molar equivalents of compound **2** in the presence of eight molar equivalents of NMI and tertiary amine (*N,N*-diisopropylethylamine) in methylene chloride, which was carried out for 24 h at room temperature (Scheme 1).

The structure of the obtained product **4** was elucidated by ^1^H and ^13^C NMR data. This product was assigned as the propargyl carbamate. According to ^13^C NMR spectra, the acetylene carbon (77.73, 78.47 ppm) and carbamate carbon signals (165.61 ppm) were still present, indicating that the acylation of amines took place. This compound has been used as a scaffold for the synthesis of multivalent carbohydrate conjugates (Scheme 2).

Methotrexate intermediate derivative **4** is an unstable compound that is practically insoluble in water and typical solvents applied in the copper-catalyzed azide−alkyne cycloaddition reaction (CuAAC) [17,18,24]. For this reason, a series of experiments adjusting the CuAAC reaction conditions were performed (Table 2).

The conventional Cu(I)-catalyzed coupling of **4** with glucosyl azide **1** afforded the desired product in low yields (11%). The low yield of glycoconjugate induced us to search for a better procedure of conjugation. As it turned out, copper(I)-catalyzed cycloaddition exclusively transformed substrates into the 1,2,3-triazole conjugate with the best yield (77%) by the treatment of sugar azide **1** with a solution of MTX carbamate **4** in the mixture of THF/*i*-PrOH/H_2_O with the addition of *N,N*-diisopropylethylamine under sonification (Scheme 2).

Glycoconjugate **5** was purified by column chromatography, and its structure was elucidated by ^1^H and ^13^C NMR data and mass spectrometry analysis. The NMR analyses were in full agreement with the expected structure (see the Appendix A). ^1^H NMR and coupling constants unambiguously confirmed β-configurations at the anomeric centers of the pyranose units. The remarkably well-resolved NMR peaks indicated that the glycoconjugate exhibited only marginal aggregation behavior in the DMSO solution.

Additionally, we have performed a mass spectrometry analysis, which indicated that within 4 h after the administration of the conjugate to the culture medium, the compound did not undergo degradation and entered the cells and released the cytotoxic payload (MTX) (Appendix A). However, we did not find other high molecular weight products (such as triazole or the linker) in the intracellular compartment, which suggests that the sugar moiety of the conjugate is quickly degraded after the internalization of the compound.

### 2.2. Biology Experiments

#### 2.2.1. Glu-MTX Compared to MTX Exerted More Potent Cytotoxic Activity on MCF-7 and SW480 Cancer Cell Lines in Hypoxia and Glucose-Deprived Microenvironment

To investigate the effect of hypoxia on the MTX chemotherapy sensitivity, MCF-7 and SW480 cells in a controlled normoxia environment and a controlled hypoxia environment were relatively exposed to the various concentrations of MTX (5~20 μM) for 48 h. Compared with MCF-7/normoxia, the cell viability (%) in MCF-7/hypoxia decreases by 30% at the MTX dosage of 20 μM (Figure 2A). The same was observed for SW480 cells, where the viability (%) in normoxic cells was 29% lower than in hypoxic cells (Figure 2B). These findings indicate that hypoxia induces MTX chemoresistance. To investigate whether glucose and methotrexate conjugation could overcome hypoxia-induced drug resistance, we treated MCF-7 and SW480 cells in a controlled hypoxia environment with Glu-MTX for 48 h. Glu-MTX exerted a significantly stronger cytotoxic effect compared to MTX in hypoxic conditions. The IC_50_ of Glu-MTX was ~10 μM in both cell lines (Figure 2A,B). Compared to MTX, Glu-MTX at a dose of 20 μM exerted a 2.3 times greater cytotoxic effect on cancer cells in a hypoxic microenvironment.

To assess the effect of Glu-MTX in glucose-deprived cells, which were slightly more resistant to MTX than regular cells, we treated MCF-7 and SW480 cells in a controlled glucose-deprived environment with MTX and Glu-MTX for 48 h. The effect of glucose starvation on the MTX chemotherapy sensitivity was assessed in MCF-7 and SW480 cells in a controlled glucose-rich environment and controlled glucose-deprived environment. The cells were exposed to various concentrations of MTX (5~20 μM) for 48 h. Compared with MCF-7 cultured with glucose medium, the cell viability (%) of MCF-7 without glucose decreases by 17% at the Glu-MTX dosage of 20 μM (Figure 2C), whereas in SW480 cells, the viability (%) in glucose-rich cells was nearly 24% higher than in glucose-deprived cells at the Glu-MTX dosage of 20 μM (Figure 2D). These findings prove that glucose starvation affects the susceptibility of cancer cells to Glu-MTX and indicate the greater efficacy of Glu-MTX compared to MTX in glucose-deprived conditions.

To examine the effect of Glu-MTX in the tumor microenvironment, which comprises hypoxia and glucose-deprived medium, we treated MCF-7 and SW480 cells in the controlled hypoxia and glucose-deprived environment with MTX and Glu-MTX for 48 h. Both cancer cell lines were resistant to MTX (cell viability 74% for MCF-7 and 80% for SW480) even at a high dose of 20 μM, whereas, at the same dose, the cytotoxic effect of Glu-MTX was significantly higher (cell viability 33% for MCF-7 and 23% for SW480). The IC_50_ of Glu-MTX in the tumor microenvironment was in the range of 4~4.5 μM for both cell lines (Figure 2E).

#### 2.2.2. Glu-MTX Inhibits the Wound Healing Process

Since the hypoxic breast and colon cancer cells could have altered migration behavior in response to different cytotoxic agents, we furthermore examined the effect of MTX and Glu-MTX on these parameters using in vitro wound-healing assay. The assay examines the migration of cells by evaluating the closure of a standard scratch in time. In both cancer cell lines, we found that Glu-MTX-treated cells had significantly slower migration than MTX-treated cells. At 48 h, the wound was unclosed in Glu-MTX-treated cells, whereas in MTX-treated cells, the gap was considerably smaller (Figure 3A,B).

#### 2.2.3. Glu-MTX Induces Apoptosis by Increasing the Expression of Caspase-3 and Bax

The expression of proapoptotic proteins—caspase 3 and Bax in SW480/hypoxia cells—was analyzed by immunocytochemistry (Figure 4). We detected high levels of proapoptotic proteins (bax and caspase-3) in cells treated with Glu-MTX and MTX compared to the control. The intensity and percentage positivity of staining were similar or slightly elevated in Glu-MTX-treated cells compared to in MTX-treated cells.

The flow cytometry analysis results are shown in Figure 5. The flow cytometry analysis results are shown in Figure 5. In comparison to the SW80 cells treated with MTX, Glu-MTX in hypoxic conditions presents increased late apoptosis after the proposed treatment. We have observed increased effectiveness of early and late apoptosis induction in almost 25% of cells after treatment with Glu-MTX (44%) in comparison to free MTX (19%). Control cells and MTX-treated cells demonstrate a similar rate of apoptosis induction and reveal decreased MTX potency in hypoxia.

## 3. Discussion

Cancer cells’ ability to evade or to handle the presence of chemotherapeutic agents is a fundamental challenge that oncology research aims to elucidate and overcome. Chemoresistance and malignant progression are closely linked with the tumor microenvironment, contributing to tumors’ response to different therapeutic modalities [31,32]. The cancer microenvironment comprises many components, including extracellular matrix proteins, cancer-associated cells, and an aberrant vasculature. These physical elements give rise to the distinctive environmental properties of hypoxia and nutrient stress that inhibit the effect of cancer treatments [32]. The undesirable impact of hypoxia on malignancies relative to radio- and chemotherapy effectiveness has been established for several decades, and the survival rate of cancer patients with severely hypoxic tumors is shorter than that of patients with normoxic tumors [33].

Our study used a controlled hypoxia condition to assess the effectiveness of methotrexate on breast and colon cancer cells. We found that hypoxic cells were significantly more resistant to methotrexate than normoxic cells. These findings are consistent with previously published studies that found that hypoxia increased the resistance to methotrexate in various cancer types, including breast cancer, melanoma, and leukemia [34,35,36]. A study by Li et al. indicated that the HIF-1α-mediated pathway played a critical role in the susceptibility of MCF-7 breast cancer cells to methotrexate [34]. Due to the increasing evidence suggesting the role of nutrient stress, particularly glucose deprivation, in tumor cell survival, angiogenesis, and drug resistance, we assessed the effectiveness of methotrexate on glucose-deprived cancer cells [37,38,39]. We found that glucose-deficient cells were marginally less susceptible to methotrexate than regular cells. 

We hypothesized that the linking of methotrexate to glucose could overcome the hypoxia- and/or nutrient stress-induced chemoresistance. The novel glycoconjugate of glucose and methotrexate exerted a strong and dose-dependent cytotoxic effect on hypoxic breast and colon cancer cells, which was nearly nine-fold more potent than MTX. Similarly, in a controlled glucose-deficient state, Glu-MTX displayed up to a three-fold enhanced cytotoxic effect in both cancer cell lines compared to MTX. As cancer cells consume glucose rampantly and given that glycolysis generates energy inefficiently, we hypothesize that Glu-MTX treatment under glucose starvation resulted in the potentiated cellular accumulation of the drug and thus resulted in an enhancement of cytotoxicity in tumor cells. 

We cannot unequivocally state that the conjugate enters the cell specifically via the GLUT 1 transporter; however, given that the glycoconjugates are highly hydrophilic, they are unlikely to be internalized via passive diffusion through the lipid cell membrane. The accumulating evidence suggests that the cellular uptake of glycoconjugates must be mediated by transmembrane transporters. GLUTs are the most frequent transporters facilitating the recognition and internalization of glycoconjugates. However, we must emphasize that the transport capability of GLUTs can be slightly influenced by complex factors, such as the structure and the substitution position of the carbohydrate, the length and steric hindrance of linkers, and the property of the payload. The past evidence suggests that the cellular uptake of some glycoconjugates is not mediated solely by GLUT but also by other receptors such as OCT2 [40]. Moreover, the potential role of other transporters such as SGLT, SWEET, and ASGPR (asialoglycoprotein receptor) should also be considered [41].

In the present study, we found an intriguing fact that in the tumor microenvironment, comprising hypoxia and glucose deprivation, MTX did not exert a cytotoxic effect on cancer cells. However, the conjugation of glucose to methotrexate allowed us to overcome the resistance and to achieve a potent, dose-dependent anticancer effect. Moreover, our studies revealed that the Glu-MTX inhibited hypoxic cells’ cell migration process more effectively than MTX did. The results confirmed that Glu-MTX-treated cells expressed high levels of antiapoptotic proteins and underwent apoptosis. 

Mounting evidence suggests that the expression of GLUTs is upregulated in many cancers under hypoxia and nutrient stress by key pro-survival pathways, including the HIF and AMP-activated protein kinase (AMPK) pathways [42,43]. This may explain our findings that Glu-MTX was significantly more potent in a hypoxic and/or glucose-deprived environment than MTX.

## 4. Materials and Methods

### 4.1. Chemistry

NMR spectra were recorded on an Agilent spectrometer 400 MHz (Agilent Technologies, Santa Clara, CA, USA) using TMS as an internal standard and CDCl_3_ or DMSO as a solvent. NMR solvents were purchased from ACROS Organics (Geel, Belgium). Chemical shifts (δ) are expressed in ppm and coupling constants (*J*) in Hz. Optical rotations were measured with a JASCO 2000 polarimeter (JASCO Corporation, Tokyo, Japan) using a sodium lamp (589.3 nm) at room temperature. Melting point measurements were performed on a Stanford Research Systems OptiMelt (MPA 100) (Stanford Research System, Sunnyvale, CA, USA). Electrospray-ionization mass spectrometry was performed on a Xevo G2 Q-TOF mass spectrometer (Waters Chromatography, Etten-Leur, The Netherlands). Reactions were monitored by TLC on precoated plates of silica gel 60 F254 (Merck, Darmstadt, Deutschland). TLC plates were inspected under UV light (λ = 254 nm) or charring after spraying with 10% sulfuric acid in ethanol. Crude products were purified using column chromatography performed on silica gel 60 (Fluka, Honeywell, NJ, USA) developed with toluene/EtOAc and CHCl_3_/MeOH as solvent systems. Organic solvents were evaporated on a rotary evaporator under diminished pressure at 40 °C.

All of the chemicals used in the experiments were purchased from Sigma-Aldrich (Saint Louis, MO, USA), ACROS Organics (Geel, Belgium), and Avantor Performance Materials Poland S.A (Gliwice, Poland) and were used without purification. Methotrexate **3**, propargyl chloroformate **2**, 2-bromoethanol, and d-glucose are commercially available. 1,2,3,4,6-Penta-*O*-acetyl-β-d-glucopyranose [44], 2-bromoethyl 2,3,4,6-tetra-*O*-acetyl-β-d-glucopyranoside [26], 2-azidoethyl 2,3,4,6-tetra-*O*-acetyl-β-d-glucopyranoside [25], and 2-azidoethyl β-d-glucopyranoside **1** [25,27] were prepared according to the respective published procedures.

#### 4.1.1. Synthesis of Carbamate **4**

Methotrexate **3** (227 mg, 0.5 mmol), *N*-methylimidazole (320 µL, 4 mmol), and *N,N*-diisopropylethylamine (165 µL, 1 mmol) were sonicated for 30 min. The mixture was cooled in ice water, and the solution of propargyl chloroformate **2** (195 µL, 2 mmol) in methylene chloride (1 mL) was added. The reaction mixture was stirred for 24 h at ambient temperature and then poured into ice water. The crude product was precipitated with acetic acid, filtered, washed with water, and drying under reduced pressure left a residue that was column chromatographed to yield bis-propargyl carbamate **4** (130 mg, 42% yield): m.p. 174–176 °C; [α]_D_^23^ = −2 (*c* = 1.0, DMSO).

^1^H NMR (400 MHz, DMSO-d6): δ 1.85–2.16 (m, 2H, 2xCHMTX), 2.32 (m, 1H, CHMTX), 2.45 (m, 1H, CHMTX), 3.21 (s, 3H, CH_3_MTX), 3.52 (m, 2H, 2xCH), 3.71 (s, 2H, CH_2_), 4.35 (m, 1H, CHMTX), 4.62–4.72 (m, 2H, CH_2_), 4.80 (s, 2H, CH_2_MTX), 6.80–6.85 (m, 2H, H-PhMTX), 7.12 (s, 1H, NH), 7.28 (s, 1H, NH), 7.68–7.76 (m, 2H, H-PhMTX), 8.60 (m, 1H, NHMTX), 8.61 (s, 1H, H-7MTX).

^13^C NMR (100 MHz, DMSO-d6): δ 25.31, 29.39, 32.88, 50.89, 50.97, 51.03, 54.14, 76.93, 77.73, 78.47, 110.36, 120.41, 120.46, 128.23, 146.26, 148.39, 150.20, 152.65, 160.80, 161.99, 165.61, 165.64, 170.95, 173.01, 173.18.

#### 4.1.2. Synthesis of Glycoconjugate **5**

Carbamate **4** (62 mg, 0.1 mmol) and azidoethyl glucoside **1** (50 mg, 0.2 mmol) were dissolved in dry *i*-PrOH (2 mL), THF (2 mL) and *N,N*-diisopropylethylamine (60 µL, 0.36 mmol). The solutions of sodium ascorbate (8 mg, 0.04 mmol) in water (1 mL) and CuSO_4_·5H_2_O (5 mg, 0.02 mmol) in water (2 mL) were mixed and added to the reaction mixture and next stirred for 24 h at room temperature. The reaction mixture was filtered, the precipitate was washed with methyl alcohol, the combined filtrate was treated with acetic acid, and the crude product was separated by filtration, then washed, dried, and purified by column chromatography to yield glycoconjugate **5** (86 mg, 77% yield): m.p. 175–177 °C; [α]_D_^22^ = −20 (*c* = 1.0, DMSO).

^1^H NMR (400 MHz, DMSO-d_6_): δ 1.71–2.10 (m, 2H, 2xCHMTX), 2.19–2.39 (m, 2H, 2xCHMTX), 2.96 (dd, 2H, *J* = 7.8 Hz, *J* = 8.6 Hz, H-2_Glu_), 3.03 (dd, 2H, *J* = 9.0 Hz, *J* = 9.4 Hz, H4_Glu_), 3.09–3.55 (m, 9H, CH_3_MTX, H-3_Glu_, H-5_Glu_, H-6a_Glu_), 3.63–3.79 (m, 4H, 4xCCH), 3.68 (dd, 2H, *J* = 1.6 Hz, *J* = 11.4 Hz, H-6b_Glu_), 3.86–3.95 (m, 2H, 2xCH), 4.01–4.12 (m, 2H, 2xCH),4.22 (d, 2H, *J* = 7.8 Hz, H-1_Glu_), 4.29 (m, 1H, CHMTX), 4.55–4.61 (m, 4H, 2xCH_2_), 4.79 (bs, 2H, OH), 5.21 (s, 2H, CH_2_MTX), 6.80–6.87 (m, 2H, H-PhMTX), 7.66–7.72 (m, 2H, H-PhMTX), 7.95 (d, 1H, *J* = 7.0 Hz, NHMTX), 8.25 (s, 2H, H-5_triaz_), 8.61 (s, 1H, H-7MTX). 

^13^C NMR (100 MHz, DMSO-d_6_): δ 24.62, 30.33, 31.64, 48.55, 49.72, 54.87, 60.62, 61.05, 67.19, 69.99, 73.26, 76.54, 76.95, 102.87, 111.09, 121.49, 125.84, 128.62, 140.91, 146.75, 149.15, 150.83, 154.04, 163.78, 165.39, 165.56, 173.56, 174.26.

HRMS (ESI-TOF): calcd for C_44_H_55_N_14_O_21_Na_2_ ([M+H]^+^): *m*/*z* 1161.3462; found *m*/*z* 1161.3710.

#### 4.1.3. LC/MS Analysis

UPLC analysis was performed on a Waters HSS T3 column (1.7 µm, 1 × 50 mm) using an Acquity UPLC system (Waters, Milford, MA, USA). The mobile phase consisted of 0.1% formic acid in water (mobile phase A) and 0.1% formic acid in methanol (mobile phase B). A gradient elution at a flow of 200 µL/min was performed according to the following: 0.5 min—5% B, 2.5 min—35% B, 3.5 min—90% B, 4.5 min—90% B, 4.55 min—5% B. The total run time was 6 min. The column temperature and the autosampler temperature were kept at 45 °C and 5 °C, respectively.

Mass spectral ionization and acquisition parameters were optimized on the Xevo G2 Q-TOF MS using electrospray ionization (ESI) in the positive ionization mode. The spray voltage, source temperature, and desolvation temperature were set at 0.5 kV, 120 °C, and 450 °C, respectively. Nitrogen was used as a desolvation and nebulizer gas. The desolvation gas flow was set at 800 L/h, and the cone gas flow was 70 L/h. Data were acquired using the Masslynx software (version 4.0, Waters, Milford, MA, USA).

### 4.2. Biology

#### 4.2.1. Cell Culture

The cell line, human colon adenocarcinoma SW40 and human breast carcinoma MCF-7 obtained from the Leibniz Institute DSMZ-German Collection of Microorganisms and Cell Cultures (DSMZ, Germany), were grown in RPMI 1640 medium and supplemented with 10% fetal bovine serum (FBS), 100 U/mL penicillin, 100 lg/mL streptomycin in a humidified incubator with 5% CO_2_ at 37 °C. The culture medium was renewed every 2–3 days. Cell culture media, trypsin, FBS, and antibiotics were purchased from Gibco (Thermo Fisher Scientific Inc., Waltham, MA, USA).

#### 4.2.2. Experiment Conditions and Cell Viability MTT Assay

Following MTT experiments, control cells were maintained in normoxia (21% O_2_, 5% CO_2_) and cultured in a complete medium with glucose. Hypoxic conditions were achieved by incubating cells in 1% O_2_, 5% CO_2_ incubator (New Brunswick Galaxy 48R, Eppendorf, Hamburg, Germany) and cultured in medium without glucose (RPMI 1640, no glucose, cat. no. 11879020).

For experiments with methotrexate and glucose conjugated MTX, cells were seeded in 96-well plates (8 × 10^3^ cells/well). The next day, the cells were treated with culture medium (control) and different doses of the compounds (5, 10, 20 µM) for 48 h.

Following incubation, an MTT assay was performed. Cell viability was determined by the ability of the mitochondrial enzyme succinate dehydrogenase to convert the yellow tetrazolium salt (MTT) into violet formazan crystals in active cells. After 4 h of incubation, the medium was removed, and the water-insoluble dye was dissolved by dimethyl sulfoxide (Sigma Aldrich, Munich, Germany), generating the color, whose intensity is directly proportional to the number of viable cells. The absorbance was measured at 570 nm using the Bio-TekBioTek ELX800 multi-well reader (BioTek, Winooski, VT, USA). The percentage of viable cells (VC) was calculated as VC (100%) = (A of experimental group/A of the control group) × 100%. MTT experiments were repeated three times, and the figures represent the mean. For further experiments, the concentration of 10 µM MTX and Glu-MTX was used to evaluate the motility ability and apoptosis by flow cytometry and immunocytochemistry.

#### 4.2.3. Wound-Healing Assay

To analyze the migration properties of MCF-7 and SW480 cells, a wound-healing assay was performed. Cells were seeded on 6-well plates in 1 × 10^6^/well density to form a confluent monolayer. With a 200 µL pipette tip, a linear scratch was made in each sample. The first photograph in time point 0 was taken. Then, the cells were incubated with a dose of 10 µM MTX and Glu-MTX for 48 h. After incubation, the second photograph was taken when the scratch was closed in the control cell samples without any treatment. The experiment was repeated three times.

#### 4.2.4. Flow Cytometry-Apoptosis Assay

To evaluate the apoptosis rate of cells in the tumor microenvironment, SW480 cells were seeded at a density of 3.5 × 10^5^/well in a 6-well culture plate. The next day, cells were treated with MTX or Glu-MTX at a dose of 10 µM for 48 h. After treatment, cells were washed with PBS solution, detached by using 0.25% trypsin in EDTA, centrifuged, and prepared according to the manufacturer’s instructions from Annexin V-FITC PI Apoptosis Detection Kit (Abcam, Cambridge, UK). First, cells were suspended in 500 µL of 1× Binding Buffer. Afterward, 5 µL of Annexin V (Annexin V-FITC) and 1 µL of propidium iodide (PI, 50 µg/mL) were added to each sample. The samples were incubated for 20 min in darkness at room temperature. For the evaluation of apoptosis, a BD Accuri C6 flow cytometer (Becton Dickinson, Franklin Lakes, NJ, USA) was used. Control cells, MTX, and Glu-MTX-treated samples were measured in triplicate. The obtained results were analyzed using the BD Accuri 6 Plus Software (Becton, Dickinson, NJ, USA).

#### 4.2.5. ICC Staining for Apoptosis Detection

For the immunocytochemical analysis, the apoptotic proteins (Caspase-3, Bax, Bcl-2) were evaluated. MCF-7 cells were seeded on 8 Chamber Eppendorf Cell Imaging Slides (Eppendorf, Germany) at a density of 8 × 10^4^/well. The following day, 10 µM Glu-MTX and MTX were added to the wells for 48 h of incubation. Next, the cells were washed twice in PBS and fixed in 4% formaldehyde (Polysciences, Warrington, PA, USA) for 10 min at RT. After PBS washing, cells were permeabilized and incubated in a blocking solution containing 5% Normal Donkey Serum (Abcam, Bristol, UK), 3% Bovine Serum Albumin (Sigma Aldrich, Germany), 0.05% Tween 20 (Sigma Aldrich, Germany), 0.2% Triton X-100 (Sigma Aldrich, Germany), in PBS for 1 h at 4 °C. Then the primary antibodies: anti-Bax, anti-Bcl-2, and anti-Caspase 3 (Abcam, UK), in dilution 1:100, were applied, and the cells were incubated overnight at 4 °C. The next day, after PBS washing, the cells were incubated with a secondary anti-rabbit antibody (Sigma Aldrich, Germany) at RT for 1 h. After PBS washing, sections were stained with 3,3′-diaminobenzidine in chromogen solution (Dako EnVision+ Dual Link System-HRP, Agilent, USA) and counterstained with Mayer’s hematoxylin for nucleus counterstaining.

The control sample was performed following the above instructions but without incubation with the compounds.

The light microscopy fitted with a digital camera (Nikon, Poland) with dry objectives 20× and 40× was used to take the photos.

## 5. Conclusions

The exploration of glycoconjugates for GLUT1-targeted cancer therapy began 25 years ago with the discovery of glufosfamide. Since then, numerous preclinical and clinical trials have been conducted on glucose conjugates in the treatment of various malignancies. 

To our knowledge, this is the first time that an evaluation of the biological activity of glucose-conjugate in the tumor microenvironment has been performed. As hypoxia and nutrient stress are known to upregulate GLUT expression on the cancer cell surface and given that GLUTs confer the tumor selectivity of Glu-MTX, the results of our study confirm the viability of the strategy to combat tumor microenvironment-induced drug resistance in solid malignancies through linking of anticancer compounds with glucose. It is of paramount importance to validate anticancer agents’ activity in models closer to their “native” microenvironment to improve the dismal success rates in transitioning anticancer agents from the laboratory to the clinic [45]. 

In summary, this work showed that the conjugation of methotrexate to glucose led to an increase in potency against malignant cells under hypoxia and nutrient stress. Although the finding has been confined to in vitro studies, our observations shed light on a potential therapeutic approach to overcome chemoresistance in cancer.

## Data Availability

The data presented in this study are available in this article or asscioiated Appendix A.

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
