# Peer review of "Overcoming Hypoxia-Induced Chemoresistance in Cancer Using a Novel Glycoconjugate of Methotrexate"

_pharmaceuticals, 2020, doi:10.3390/ph14010013_

Round 1

Reviewer 1 Report

The paper "Overcoming hypoxia-induced chemoresistance in cancer using novel glycoconjugate of methotrexate" by M. Wozniak et.al is the interesting paper describing the new glucose derivative of the known drug methotrexate. The idea of using glucose conjugates in order to increase drug transport to tumors is not entirely new, however, the authors uses the knowledge about the importance of the position of the substituted drug on glucose transport through the GLUT 1 transporter. For this reason, the work is interesting and important, especially that the results obtained (in contrast to the majority of results obtained by other research groups) show the high biological activity of the derivative.

The description of the synthesis and chemical methodology is at a very high level. The biological characteristics of the conjugate are slightly worse. Have the authors found evidence that the conjugate enters into the cell via the GLUT 1 transporter and not due to partial degradation in the culture medium and free diffusion of the degradation product? Can the degradation process of the conjugate and the fate of the conjugate in the cell be predicted based on the chemical structure? This is particularly important due to the fact that two glucose molecules are conjugated with methotrexate. This is a new approach and although it has produced good results, it needs to be proved or at least well discussed.
Do the authors have data showing the rate of conjugate breakdown in the culture medium? It would be an interesting and important supplement to the obtained results.

Beside these comments, the results obtained are valuable and deserve publication.

Less important notes:

Page 3 line 117 instead of stable is: "sTable 4C1"
Page 7 line 236. The term "annihilation" is not very biological. Please correct.

Author Response

Thank you for inviting us to submit a revised draft of our manuscript entitled “Overcoming hypoxia-induced chemoresistance in cancer using novel glycoconjugate of methotrexate” to Pharmaceuticals. We also appreciate the time and effort you and each of the reviewers have dedicated to providing insightful feedback on ways to strengthen our paper. Thus, it is with great pleasure that we resubmit our article for further consideration. We have incorporated changes that reflect the detailed suggestions you have graciously provided. We also hope that our edits and the responses we provide below satisfactorily address all the issues and concerns you and the reviewers have noted. All changes are highlighted in the manuscript.

To facilitate your review of our revisions, the following is a point-by-point response to the questions and comments delivered in the review report.

Reviewer 2 Report

The article reports on a glucose methotrexate conjugate to overcome tumor microenvironment resistance to chemotherapeutic drug in cancer cells and improve their cytotoxicity.

The data are in vitro only and the work can be considered a preliminary report to be validated.

Particularly, my concern in in vivo model is the target specificity considering the ubiquitous character of glucose uptake (particularly in muscle cells) and potential binding to lectins in different tissues.

I suggest to comment more on these aspects in the conclusions.

Additional suggested changes:

In page 4-5. the chemical part is quite lengthy considering the moderate complexity of the synthesis. Particularly azidoethyl glucosides are compounds prepared with quite standard carbohydrate chemistry protocols. I am surprised of such detailed discussion. I would add some reference (see for instance Carbohydrate Research, 1984, 125, 231-245;  and the patent US20110059477"US20110059477A1) and shorten this paragraph. Also, I recommend to reduce and simplify the synthetic schemes. I believe this is not the focus of this paper.

Page 8 Figure 2. Are data on in vitro evaluation coming from one single experiment or more? this should be clearly stated in the legend. How the authors established whether differences are significant? are data glucose dependent? have authors thought to use another sugar-conjugate as control. what is the target specificity of the conjugate since glucose is very likely recognized by various lectins or uptaken by cells other than cancer ones? 

The authors should consider biodistribution vs glucose in animal model? would it be sequestered in place of glucose?

Author Response

(The authors gave the same response as above.)

Reviewer 3 Report

In their manuscript, Agarwal et. al describe the development and applications of a novel glycoconjugate based on methotrexate. The authors conduct a relatively known chemical synthesis to gain access to the bis-glycosidic-drug conjugate which is then tested for its activity under hypoxia. Possibly the efficient GLUT-uptake as previously known in the case of glycoconjugates assists in the activity of compound 9 compared to the free drug. The bigger picture of this work is to evaluate the activity of a glycoconjugate in the tumor microenvironment. This study aims to contribute to the field a technology that can overcome hypoxia-induced drug resistance specifically overcome the MTX therapy-related drawbacks.

While glycoconjugates have, in numerous reports, been utilized to improve the overall therapeutic index of MTX, their use to overcome hypoxia-driven chemoresistance is an addition to the field. However, broadly speaking, the originality of this work is not particularly significant.  The writing section of the paper needs significant improvement. The reviewer also believes that not demonstrating that the cytotoxin is indeed released in its free form is one of the important shortcomings of this manuscript (since the payload with the linker handle is also enabled to show biological activity). However, the conclusions are in line with the experimental outcomes and the reviewer believes that this work is merited to be reconsidered for publication after necessary revisions.

Additional comments:

Line 17: Authors are suggested to change the cells "rising" to growing.

Line 23: Fix the grammatical error in the sentence.

Line 82: Use of novel and the first time in the same sentence is redundant.

Line 122: As a general comment, the authors seem to refer to the carbamate bond as the amide/ester bond interchangeably (e.g. line 193). This needs to be addressed and changed to carbamate.

Line 126 is essentially commenting on the concept of prodrugs, the authors are requested to remove any unnecessary text that might be diverting the focus of the manuscript.

Line 151: Fix error in boron trifluoride "di"-ethyl etherate.

Author Response

(The authors gave the same response as above.)

Round 2

Reviewer 3 Report

Most of my comments have been addressed and I am happy to publish this manuscript in its present form.